# Randomized Controlled Trial of Simple Salt Reduction Instructions by Physician for Patients with Type 2 Diabetes Consuming Excessive Salt

**DOI:** 10.3390/ijerph18136913

**Published:** 2021-06-28

**Authors:** Chikako Oyabu, Emi Ushigome, Yuriko Ono, Ayaka Kobayashi, Yoshitaka Hashimoto, Ryosuke Sakai, Hiroya Iwase, Hiroshi Okada, Isao Yokota, Toru Tanaka, Michiaki Fukui

**Affiliations:** 1Department of Endocrinology and Metabolism, Kyoto First Red Cross Hospital, Kyoto 605-0981, Japan; Iwase@koto.kpu-m.ac.jp (H.I.); toru-tanaka@kyoto1-jrc.org (T.T.); 2Department of Endocrinology and Metabolism, Graduate School of Medical Science, Kyoto Prefectural University of Medicine, Kyoto 602-8566, Japan; emis@koto.kpu-m.ac.jp (E.U.); u_lic0@icloud.com (Y.O.); y-hashi@koto.kpu-m.ac.jp (Y.H.); michiaki@koto.kpu-m.ac.jp (M.F.); 3Department of Diabetes Medicine, Fukuchiyama City Hospital, Kyoto 620-8505, Japan; a0911@koto.kpu-m.ac.jp; 4Department of Diabetes Medicine, Kyoto City Hospital, Kyoto 604-8845, Japan; c030eb@koto.kpu-m.ac.jp; 5Department of Diabetes and Endocrinology, Matsushita Memorial Hospital, Moriguchi 570-8540, Japan; conti@koto.kpu-m.ac.jp; 6Department of Biostatistics, Graduate School of Medicine, Hokkaido University, Sapporo 060-0808, Japan; yokotai@med.hokudai.ac.jp

**Keywords:** salt intake, salt restriction, nutritional instructions, randomized controlled trial, hypertension, type 2 diabetes mellitus

## Abstract

*Objectives*: We verified the clinical usefulness of an approach method in which a physician gives simple salt reduction instructions during outpatient visits to patients with type 2 diabetes. *Methods*: This study was an open-blind, randomized controlled trial. Subjects were outpatients with type 2 diabetes whose estimated salt intake using spot morning urine sample exceeded the target of salt intake. The control group (CG) was notified only of the current salt intake, whereas the intervention group (IG) was given the brief salt reduction instruction by a physician in addition to the information regarding their current salt intake. *Results*: The change in estimated salt intake was −0.6 g (from 10.1 to 9.5 g, *p* = 0.029) in the CG after 8 weeks, and −0.9 g (from 10.1 to 9.2 g, *p* = 0.001) in the IG, although there were no significant differences between them (*p* = 0.47). After 24 weeks, both groups no longer differed significantly from the baseline. In addition, multivariate linear regression analyses indicated that high salt intake and low estimated glomerular filtration rate at baseline were significantly associated with salt reduction after 8 weeks. *Conclusions*: Salt-reducing effects were observed after 8 weeks in both the IG and CG, but no significant difference was observed. Moreover, patients with high salt intake and renal disfunction may be more effective in accepting salt reduction instructions. Making patients aware of the importance of salt reduction through a physician is effective for continuous salt reduction, and it is important to continue regular and repetitive guidance.

## 1. Introduction

It is well known that excessive salt intake is directly associated with an increased risk of cardiovascular disease and cardiovascular mortality [1,2]. When glycemic control in patients with type 2 diabetes is inadequate, this association becomes synergistically stronger [3]. Furthermore, the comorbidity of diabetes with hypertension significantly increases the incidence of cardiovascular disease [4,5]; therefore, in addition to glycemic control, intensive treatment for hypertension is essential for patients with diabetes. In Japan, salt intake is excessive compared to other countries, and sufficient restrictions on dietary sodium intake have not yet been implemented [6]. In addition, it has been reported that approximately half of the Japanese population is highly sensitive to salt, suggesting that salt reduction may be useful to the medical economy [7]. The effects of reduced salt guidance on both healthy and hypertensive populations have previously been reported [8,9], but no studies have examined the effect of reduced salt guidance specifically in patients with diabetes.

In a previous single-arm trial, we reported that a national registered dietitian’s guidance on salt reduction is beneficial for blood pressure management at home and for lowering the salt intake in patients with type 2 diabetes who consume it in excess [10]. In that study, daily salt intake was significantly reduced to 0.8 g/day and 0.7 g/day at 2 months and 6 months after guidance, respectively. In daily clinical practice, comprehensive salt reduction guidance by registered dietitians is widely recommended. However, guidance is usually provided by appointment, and instruction can be time-consuming. Hence, it is difficult for all patients with type 2 diabetes and excessive salt intake, who are regularly seen in daily clinical practice, to receive such guidance. Therefore, it is of great clinical significance if physicians can provide concise nutritional instruction for outpatients with diabetes during regular medical treatments and prove the instruction is effective. In this study, we hoped to verify the clinical usefulness of simple salt reduction instructions given by physicians to outpatients with type 2 diabetes, thus providing efficient salt reduction guidance for a wider range of patients and contributing to the prevention of the progression of hypertensive diseases.

## 2. Materials and Methods

### 2.1. Study Design

The design of this study has been reported previously [11]. Briefly, this was a non-blinded randomized controlled trial. We extracted the data of type 2 patients with diabetes with excessive salt intake who were treated at one of two diabetes clinics (in Kyoto Prefectural University of Medicine Hospital and Kyoto First Red Cross Hospital) and assigned them to either the intervention group (IG) or the control group (CG).

### 2.2. Patients

Between April 2016 and December 2018, we recruited 260 patients with type 2 diabetes who had regularly attended the diabetes outpatient clinics. Eligible patients were:(1)Adults aged 20 to 90 years old who had been diagnosed with type 2 diabetes;(2)Those with an estimated salt intake ≥8 g/day for males, ≥7 g/day for females, or ≥6 g/day for patients with hypertension, as determined through a morning urinalysis during the outpatient visit [12,13].

Patients were excluded if they met any of the following criteria:(1)Unstable dietary intake;(2)Secondary hypertension or malignant hypertension;(3)Pregnant, breastfeeding, or planning to become pregnant;(4)Hospitalization due to serious heart disease;(5)Severe cerebrovascular accident;(6)Advanced renal failure (serum creatinine ≥ 2.0 mg/dL or patient currently on dialysis);(7)Having a general condition exacerbated by malignant disease;(8)Severe psychiatric disorders or severe neuropathy, including epilepsy;(9)For other safety reasons, the principal investigator or the sub-investigator determined whether participation in this study would have been inappropriate.

### 2.3. Randomization

All participants were randomized into two arms, intervention or control, using a size 10 substitution block method. Immediately after obtaining informed consent from the participants, another researcher, independent of the attending physician, randomly assigned participants to each group. The patient and their attending physician were informed as to whether they were assigned to the intervention group (IG) or control group (CG). Due to the nature of this study, participants, physicians, and some researchers could not be blinded to the presence or absence of the intervention.

### 2.4. Procedure

At the start, all patients were informed and understood that they were taking excess salt and the importance of salt reduction. Patients in the CG were informed of their current daily salt intake and recommended salt intake levels by their physician during the baseline outpatient visit, but did not receive any special short-term salt reduction instructions. During the subsequent outpatient visits, normal medical care was continued. Participants assigned to the IG on the study enrollment date were notified of their current salt intake, as in the CG, and then received a brief salt reduction instruction from their physician for approximately 5 min only at baseline. The content of the brief guidance mainly included the “12 tips for salt reduction”. The twelve tips on reducing salt intake are: 1. Become used to more mild flavors; 2. Limit the intake of pickles and soup; 3. Bring out the effect with a little salt; 4. Dip, do not sprinkle, food in salty sauces such as soy sauce; 5. Make good use of sour-tasting ingredients; 6. Use spices abundantly; 7. Use ingredients with natural scents such as herbs and perilla; 8. Roasted aroma is your friend (sesame, nuts, fish, vegetable, etc.); 9. Use flavorful oil such as sesame oil, olive oil, and herbal oil; 10. Limit the intake of snack-eating with alcohol; 11. Limit the intake of fish paste products and processed foods; 12. Do not overeat. In addition, as reference materials, “Specific examples of the amount of seasoning equivalent to 1 g of salt”, “Specific examples of the estimated amount of salt contained in foods (such as dried foods, pastes, processed foods, ready-to-eat foods, pickles, delicacies,)”, and “Specific examples of the estimated salt intake while eating out (such as noodles, bowls, light meals, dim sum, set meals)” were handed to the patient. Patients (both the IG and CG) were then informed of their current estimated daily salt intake at 8 and 24 weeks of follow-up. Patients in the CG did not receive intensive instructions with the above pamphlets; the guidance provided to these patients was similar to that of regular clinic visits regarding general healthy eating.

### 2.5. Primary Outcome

The primary outcome was daily salt intake and was assessed at baseline, 8 weeks, and 24 weeks. Considering that it can be measured clinically with ease, the estimated salt intake was measured using a spot urine sample in the morning and calculated using the following formula: daily salt intake (g/day) = 0.0585 × 21.98 × {urinary sodium (mEq/L)/urinary creatinine (mg/dL) × (14.89 × body weight (kg) + 16.14 × height (cm) − 2.04 × age (year) − 2244.45)}0.392 [14]. The spot urine sample was collected by an immunoturbidimetric assay (Autokit Micro Albumin, Wako, Osaka, Japan).

### 2.6. Secondary Outcomes

Secondary outcomes were body weight, body mass index, blood pressure, HbA1c, lipid profile (triglyceride, HDL cholesterol), creatinine, estimated glomerular filtration rate (eGFR), and urinary microalbumin excretion (UAE). Biochemical blood tests were performed in the morning. HbA1c, serum lipid profiles, and other biochemical data were measured in standard laboratories. UAE was measured by immunoturbidimetric analysis. HbA1c was expressed using the National Glycohemoglobin Standardization Program units. Data on sex, age, history of diabetes, smoking/ drinking status, and medication history were obtained by asking individual participants upon initiation of the study. Regarding complications from diabetes, retinopathy from diabetes was evaluated from the description in the medical record. Nephropathy was evaluated by classifying UAE into the following three categories: normoalbuminuria: UAE < 30 mg/g Cr; microalbuminuria: 30–300 mg/g Cr; or macroalbuminuria: >300 mg/g Cr). Furthermore, neuropathy was evaluated based on the diagnostic criteria for polyneuropathy from diabetes proposed by the Diabetic Neuropathy Research Group [15]. The presence or absence of macrovascular complications was determined based on medical history or physical examination, by defining it as a history of cardiovascular disease, cerebrovascular disease, or arteriosclerosis obliterans. Patients who had switched, discontinued, or started new hypoglycemic or antihypertensive medications during the study period were excluded from the analyses. Patients who were newly prescribed or discontinued thiazide diuretics and sodium glucose cotransporter 2 inhibitors involved in urinary sodium excretion were also excluded from the analyses. In principle, the outcomes were not changed after trial commencement.

### 2.7. Statistical Methods

A mixed model was used to analyze factors related to salt intake in patients with diabetes. Factors that might affect salt intake and their changes were adjusted for a multivariate model (a decrease in estimated salt intake of 0.1 g or more was defined as an improvement). Subgroup analyses were performed by age (<65 or ≥65), sex (male or female), and HbA1c level (˂7% or ≥7%), with or without antihypertensive agents, and the difference in instruction effects between subgroups. JMP version 11.2.0 software (SAS Institute, Cary, NC, USA) was used for statistical analyses. We calculated the mean or frequency of evaluation items, expressed categorical variables as percentages (numerical values), and continuous variables as mean (SD). If the data had missing values, the multiple assignment method was applied. *p* < 0.05 was considered statistically significant. The difference between the two groups was shown by the 95% confidence interval and the *p*-value. Previous studies reported that salt intake decreased from 10.3 ± 1.6 g to 9.3 ± 1.9 g/day after salt reduction guidance for type 2 diabetes [10]. From this result, the sample size was calculated based on the assumption that the daily salt intake would be reduced by 1 g in the IG and the intake would not change in the CG.

If the bilateral significance level were set to 5.0% and the detection power were set to 80%, the required sample size would be 90 participants in each group. Assuming a 10% dropout rate, a total of 200 participants were required for both groups.

This study has been approved by the research ethics committee (or institutional review board) of DEF University.

## 3. Results

We assigned 132 patients who participated in our study to the IG and 128 to the CG. In total, 250 participants were analyzed, excluding those whose estimated salt intake data was missing after registration. Table 1 shows the clinical characteristics data. Estimated baseline salt intake was 10.1 g in both groups. Changes in estimated salt intake were evaluated using a mixed model to see if there was a significant difference between the IG and CG. After 8 weeks (Figure 1), the CG showed a decrease of 0.6 g (from 10.1 g to 9.5 g, *p* = 0.029), and the IG showed a decrease of 0.9 g (from 10.1 g to 9.2 g, *p* = 0.001). All patients showed significant changes compared to the baseline. After 24 weeks, CG had a decrease of −0.2 g (from 10.1 g to 9.9 g, *p* = 0.529) compared to baseline, and IG had a decrease of −0.4 g (from 10.1 g to 9.7 g, *p* = 0.217) from baseline, but both groups no longer differed significantly from the baseline. Salt intake during the 24 weeks was not different between the two groups.

The results of the secondary outcomes are shown in Figure 2. Regarding blood pressure, a significant decrease of −2.5 mmHg compared to baseline in systolic blood pressure was observed in the IG after 8 weeks. In addition, the CG showed a significant increase of 0.1% compared to baseline in HbA1c after 24 weeks. No significant changes were observed in renal function, liver function, lipid profile, or UAE after 8 and 24 weeks.

The subgroup analysis by sex showed that the salt intake after 8 weeks was significantly decreased in men assigned to the IG (from 10.5 g to 9.3 g, *p* = 0.0005). In addition, when patients were divided into two groups according to age (≤64 years or ≥65 years), salt intake was significantly reduced after 8 weeks in the ≥65-year-old patients of both groups (IG: from 10.0 g to 8.9 g, *p* = 0.0005; CG: from 10.0 g to 9.3 g, *p* = 0.0275). Furthermore, when the group with good glycemic control and the group with poor glycemic control (HbA1c < 7.5% or HbA1c ≥ 7.5%) were compared, a significantly improved effect (from 10.0 g to 9.3 g, *p* = 0.0072) was observed in the IG with good glycemic control. In addition, when patients were divided into two groups according to BMI (BMI < 25 or BMI ≥ 25), salt intake was significantly reduced after 8 weeks in patients with BMI < 25 (control group 10.0 g to 9.5 g, *p* = 0.009, intervention group 9.8 g to 8.9 g, *p* = 0.005), and those with BMI ≥ 25 (control group 10.3 g to 9.4 g, *p* = 0.01, intervention group 10.5 g to 9.6 g, *p* = 0.006). Furthermore, the presence or absence of antihypertensive drugs was examined. In the group of patients taking antihypertensive drugs, a significant salt reduction effect (IG: from 10.0 g to 9.1 g, *p* = 0.0059, CG: from 10.1 to 9.2 g, *p* = 0.013) was recognized. Regarding antihypertensive agents, analysis was performed only on patients who were taking angiotensin II receptor blockers (ARB) or CCB. In the group taking both ARB and CCB, there was a significant salt reduction effect after 8 weeks in the IG; however, no significant effects were observed in any of the groups taking either ARB or CCB.

Finally, through a multivariate analysis (Table 2), we examined the factors that were associated with salt intake reduction after 8 weeks in patients with type 2 diabetes. We found that high salt intake and low eGFR at baseline showed a significant association. Regarding the factors associated with continuous salt intake reduction after 8 weeks and 24 weeks, only high salt intake at baseline was significantly associated.

## 4. Discussion

It is estimated that there are currently 43 million patients with hypertension in Japan, and about one in three Japanese people have hypertension [16]. East Asians, including the Japanese, have weak insulin secretory capacity, and lifestyle changes have led to an increase in the number of people with visceral fat-accumulating obesity [17,18]. These patients are in a high-risk group for macroangiopathy, resulting in poor prognoses.

In Japan, approximately half of the patients with type 2 diabetes have hypertension [19]. In particular, type 2 diabetes with insulin resistance is likely to cause salt-sensitive hypertension [20,21], and it is expected that salt reduction will be effective in suppressing cardiovascular events. In addition, it has been reported that when patients with type 2 diabetes reduce their salt intake, glucose tolerance is improved, and albuminuria is decreased [22].

In this study, we examined the effects of short-term focused instructions delivered by physicians based on the results of previous studies. The salt intake recommended by the WHO guidelines to achieve a significant reduction in blood pressure is 5 g/day. This is different from the target salt intake in this study. According to the Japan National Health and Nutrition Examination Survey 2018, the average salt intake of Japanese adults is 10.1 g. Furthermore, those who were able to consume less than 5 g/day were considered to be below the lower limit of the fifth percentile value, which was not considered appropriate as an amount that could actually reduce salt.

The results of this study showed a significant decrease in salt intake in the IG 8 weeks after baseline. However, the effect could not be sustained until the 24th week, and intake increased again; this course was similar to the findings of our previous study. Similarly, salt intake in CG decreased significantly 8 weeks after baseline and was not significantly different from the baseline at 24 weeks; this result was different from what we expected. In this study, patients in both groups were informed by their attending physicians regarding baseline salt intake, and the study aimed to reduce salt intake and to follow up on salt intake thereafter. Therefore, it is possible that a little awareness of the patient’s salt intake may have influenced the results. Making patients aware of salt reduction by a physician (or at least recognition of their excessive salt intake) is effective for continuous salt reduction, with or without instructions.

In the present study, high BMI had significantly higher baseline salt intake than low BMI (10.4 g vs. 9.8 g, *p* = 0.047), but similar effects from salt reduction guidance were observed.

High salt intake at baseline and low eGFR were associated with a salt-reducing effect after 8 weeks, and high salt intake at baseline was associated with a continuous salt-reducing effect at 8 and 24 weeks. As with previous speculation, higher-risk patients may be more likely to be receptive to guidance for low salt consumption.

In the present results, there was no improvement in HbA1c levels and fasting glucose levels at 8 weeks in both groups. It may be that the behavior toward salt reduction in this study did not lead to a reduction in carbohydrates and total calories. According to previous reports, salt reduction has been reported to improve glucose tolerance and decrease albuminuria in patients with type 2 diabetes, and if more stringent salt reduction can be achieved, changes in parameters related to diabetes can be expected.

On the other hand, only the control group showed a worsening in HbA1c after 24 weeks. The reason for this is not clear. There was no bias in the season of intervention between the intervention group and the control group. The reason for maintaining the HbA1c level in the intervention group was that some of the 12 tips used in the salt reduction instruction may have been helpful.

In this study, we performed several subgroup analyses. As a result, a significant salt-reducing effect was observed after 8 weeks in men, the elderly group, and those who took antihypertensive drugs. It is possible that, in men, the high salt intake at baseline has an impact on its reduction. Older patients have a tendency to eat out less frequently and often have a more regular schedule than younger patients, making it easier to adjust their diets. In the subgroup analysis with and without antihypertensive drugs, a significant salt-reducing effect was observed after 8 weeks in those receiving antihypertensive drugs. This might be because patients who were taking antihypertensive drugs were more likely to have renal dysfunction with reduced eGFR.

Furthermore, when the antihypertensive agents were examined separately, according to the presence or absence of RAS and CCB, a significant salt reduction was observed after 8 weeks in the IG receiving both agents (RAS and CCB). However, no significant salt-reducing effect was observed in the groups taking only one drug (RAS or CCB). It is not clear why the two-drug group was more likely to have a salt-reducing effect, but patients taking the two-drug had higher blood pressure, higher salt intake, lower eGFR, and were older, which may have increased their awareness of salt intake and blood pressure.

Regarding drug therapy, it has been reported that hydrochlorothiazide, or a combination of hydrochlorothiazide and CCB, has a superior salt-reducing effect among antihypertensive agents [23,24]. Furthermore, it has been reported that the combination of metformin and CCB is also useful in patients with obesity [24], and this treatment might be worth further investigation in the future, especially in obese patients with type 2 diabetes. However, it has also been suggested that low salt intake among patients with heart failure may result in unintended nutritional problems (undernutrition and lack of electrolytes and trace elements) [25]. It is therefore necessary to consider the whole diet, particularly when instructing elderly patients, who are prone to undernutrition, and patients with underlying medical conditions.

There are several limitations of this study. The first is that this study was a non-blinded (for both participants and physicians) randomized controlled trial. This could have consciously biased the effect on salt reduction. However, in this study, we adopted an intervention method in an environment closer to daily clinical practice; hence, it was impossible to completely blind the study. Moreover, from the results of this study, the IG tended to have a higher salt-reducing effect, but the effect was also observed in the CG. The fact that even a little awareness may lead to salt-reducing behavior is noteworthy. Next, the estimated salt intake was calculated from an early morning urine sample, which lacks accuracy. Previous studies have reported a higher reliability of salt intake measured from urine collection over 24 h. This method is considered highly reliable because it is an easy test widely used in daily clinical practice.

## 5. Conclusions

In this study, we have not shown a significant effect of simple salt reduction instructions by physician for patients with type 2 diabetes consuming excessive salt. However, it was clarified that making patients aware of the importance of salt reduction is effective for continuous salt reduction, and it is important to continue repetitive instruction. In the future, we would like to plan an approach that not only reduces salt intake, but also has a positive effect on blood pressure and renal function in patients with diabetes.

## Figures and Tables

**Figure 1 ijerph-18-06913-f001:**
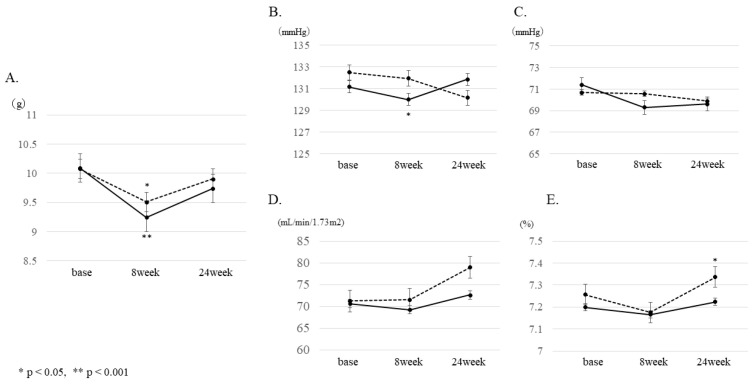
Changes at 8 and 24 weeks from baseline. Changes at 8 and 24 weeks from baseline were compared between the intervention group (IG) and control group (CG) using mixed models for (**A**) estimated salt intake, (**B**) systolic blood pressure (SBP), (**C**) diastolic blood pressure (DBP), (**D**) estimated glomerular filtration rate (eGFR), and (**E**) HbA1c. (* *p* < 0.05, ** *p* < 0.001) Solid line: intervention group (IG), dotted line: control group (CG).

**Figure 2 ijerph-18-06913-f002:**
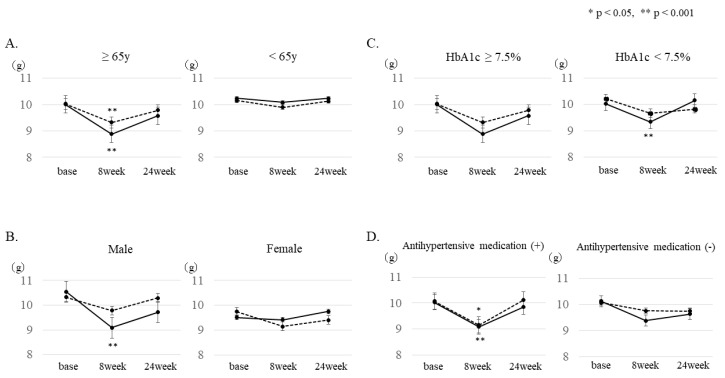
Subgroup analysis of changes in estimated salt intake. Changes in estimated salt intake 8 and 24 weeks after baseline were analyzed in the following subgroups: (**A**) age, (**B**) gender, (**C**) with or without antihypertensive drugs, and (**D**) HbA1c. A mixed model was used for the analysis. (* *p* < 0.05, ** *p* < 0.001). Solid line: IG, dotted line: CG.

**Table 1 ijerph-18-06913-t001:** Clinical characteristics of study patients.

Characteristic	Intervention *n* = 124	Control *n* = 126	*p*
Age (year)	66.4 (11.0)	67.3 (11.4)	0.507
Men (%)	55.6 (69)	56.3 (71)	0.910
Duration of diabetes (years)	15.5 (10.5)	16.7 (9.8)	0.397
Systolic blood pressure (mmHg)	131.1 (16.4)	132.6 (17.5)	0.495
Diastolic blood pressure (mmHg)	71.5 (11.9)	70.7 (11.8)	0.627
Body mass index (kg/m^2^)	23.8 (4.6)	24.4 (3.9)	0.302
HbA1c (%)	7.2 (1.1)	7.3 (0.9)	0.559
Total cholesterol (mg/dL)	191.1 (37.6)	181.8 (29.5)	0.073
Triglycerides (mg/dL)	152.9 (92.7)	139.2 (111.0)	0.298
Creatinine (mg/dL)	0.8 (0.4)	0.8 (0.3)	0.694
Estimated glomerular filtration rate (mL/min/1.73 m^2^)	70.8 (22.1)	71.5 (23.6)	0.797
Salt intake (g)	10.1 (2.2)	10.1 (2.1)	0.992
Smoking status (current/past)	20.1/34.4 (24/41)	16.9/44.0 (20/52)	0.316
Drinking status (current/past)	20.1/27.7 (22/33)	18.6/27.9 (22/33)	0.998
Usage of antihypertensive medications	44.3% (55)	40.4% (51)	0.337
Usage of RAS inhibitors	29.0% (36)	27.8% (35)	
Usage of CCBs	22..5% (28)	25.3% (32)	
Usage of Diuretics	15.3% (19)	6.3% (8)	
Nephropathy (stage1/stage2/stage3,4)	56.0/42.2/1.6 (69/52/2)	56.9/40.6/2.4 (70/50/3)	0.884
Retinopathy (NDR/SDR/PDR)	69.3/14.5/16.1 (86/18/20)	72.5/12.0/15.3 (90/15/19)	0.823
Neuropathy	36.5 (45)	37.3 (46)	0.894
Macrovascular complication	20.3 (25)	15.3 (19)	0.369

Data are expressed as mean (SD) or % (number). The difference between groups was analyzed by Student’s *t*-test or chi-squared test. RAS, renin angiotensin system; CCB, calcium channel blocker; NDR, no retinopathy from diabetes; SDR, simple retinopathy from diabetes; PDR, proliferative retinopathy from diabetes.

**Table 2 ijerph-18-06913-t002:** Odds ratios of factors associated with salt intake reduction after 8 weeks, or after 8 weeks and 24 weeks.

Characteristic	After 8 Weeks		After 8 and 24 Weeks	
	OR (95% CI)	*p*	OR (95% CI)	*p*
Duration of DM (year)	1.00 (0.981.04)	0.614	0.98 (0.92–1.03)	0.503
BMI (kg/m^2^)	0.97 (0.89–1.06)	0.471	1.02 (0.91–1.12)	0.758
HbA1_C_ (%)	0.21 (0.85–1.73)	0.282	0.77 (0.40–1.33)	0.377
eGFR (mL/min/1.73 m^2^)	0.83 (0.71–0.97)	0.010	1.00 (0.98–1.02)	0.765
SBP (mmHg)	1.001 (0.99–1.03)	0.494	1.01 (0.97–1.04)	0.639
DBP (mmHg)	1.00 (0.97–1.03)	0.951	0.98 (0.93–1.02)	0.328
Antihypertensive medication (+/−)	0.67 (0.35–1.30)	0.237	1.30 (0.49–3.63)	0.605
Instruction to reduce salt (+/−)	1.08 (0.58–2.00)	0.811	1.20 (0.49–3.03)	0.687
Salt intake at baseline (g)	1.53 (1.27–1.87)	<0.0001	1.35 (1.09–1.69)	0.005
Smoking (never vs. current)	0.48 (0.19–1.13)	0.130	2.13 (0.59–10.34)	0.558
Smoking (never vs. past)	0.80 (0.59–2.63)	0.683	2.02 (0.68–6.61)	0.579

DM: diabetes mellitus; SBP, systolic blood pressure; DBP, diastolic blood pressure.

## Data Availability

Data available on request due to restrictions eg privacy or ethical. The data presented in this study are available on request from the corresponding author. The data are not publicly available due to the protection of personal information.

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
