# Peer review of "Randomized Controlled Trial of Simple Salt Reduction Instructions by Physician for Patients with Type 2 Diabetes Consuming Excessive Salt"

_ijerph, 2021, doi:10.3390/ijerph18136913_

Round 1
Reviewer 1 Report
Line 53:
I thintk the authors must to modify 0.8 g and 0.7 g by 0.8 g/day and 0.7 g/day
Line 76:
If the WHO salt intake recommendation is 5 g/day, why have not the authors included this recommendation in their salt intake?
Line 103:
Please, could do you say whta are the 12 tips for salt reduction?
Reviewer 2 Report
Dear Authors,
I would like to ask why the Japanese population has a low capacity to secrete insulin and its possible effects on signaling and therefore these people are more prone to developing diabetes. In addition, I would like to know if the Japanese population also consumes foods with a high calorie content that favor the development of the disease. The answer to these questions can also be considered in the discussion.
Reviewer 3 Report
The article is of potential interest because it shows that a simple clinical practice such as making people aware of the relevance of salt assumption can lead to a reduction of salt assumption. However there are several points to improve.
A fundamental point of the study is that in the long term (after 24 weeks), people don't show a reduction in salt assumption. This point should be more extensively discussed.
Additionally it is not clear from the study if salt reduction improves metabolic parameters related to T2D.
I suggest a major revision concerning the following points:
1) The study is of potential interest because it shows that making patients aware of the importance of salt reduction is effective for continuous salt reduction, at least in the short term.
Unfortunately it also shows that it does not work in the long term (after 24 weeks). It would be of interest verifying if a constant reminder by clinicians concerning the importance of reducing salt in the diet would change or not the outcome after 24 weeks.
Why did you not performed the study till week 24?
2) Additionally, it is not clear if the salt reduction recorded by the authors after 8 weeks is related to the improvement of some metabolic parameters (beyond HbA1c) related to T2D. Did the author recorded basal glucose and/or basal insulin secretion? It would be of interest seeing if they are related to the decrease in salt assumption. If the authors don’t have such data, it would be important to at least mention studies showing the relevance of reducing salt concentration to improve metabolic parameters related to T2D.
3) It would be of interest to perform subgroup analysis such as that reported in figure 2 also for BMI (if the included subjects belong to different BMI classes).
4) Why HbA1c increases in control group after 24 weeks ? They don’t show significant differences from baseline or from intervention group after 24 weeks so how do you explain this result? Please discuss it.
5) Why in people with HbA1c < 7.5 % salt assumption decreases while it does not in people with HbA1c > 7.5 %. Please discuss it.
There are also some minor revisions to perform:
Page 183, line 172
“showed a of 0.9 g” should be changed into “showed a decrease of 0.9 g”
Line 173
“decrease of -0.2 g” should be changed into “decrease of 0.2 g”
Line 174
“decrease of -0.4 g“ should be changed into “decrease of 0.4 g“
Line 178
decrease of 2.5 mmHg not – 2.5 mmHg
Line 221-222
You wrote: “It is estimated that there are currently 43 million patients with hypertension in Japan, and about 1 in 3 Japanese people have hypertension.”
The two sentences state the same thing. Choose one or the other.
Round 2
Reviewer 2 Report
The authors of the manuscript heeded the suggested recommendations and broadened the discussion of their results a bit. My suggestion is to correct a redaction error on page 22 as a word is cut off and the word value2 appears. Please to correct the error.
Reviewer 3 Report
The manuscript can be accepted in the present form.